# Aqueous particle generation with a 3D printed nebulizer

Michael Rösch[1,2] , and Daniel J. Cziczo[1,3,4]

[1]Department of Earth, Atmospheric & Planetary Sciences, Massachusetts Institute of Technology,

Cambridge, 02139, USA

[2]Department of Environmental Systems Science, Eidgenössische Technische Hochschule - ETH, Zurich, 8092, Switzerland

[3]Department of Civil Environmental Engineering, Massachusetts Institute of Technology, Cambridge, 02139, USA

[4]Department of Earth, Atmospheric and Planetary Sciences, Purdue University, West Lafayette, IN 47907 USA

*Correspondence to*: Michael Roesch (michael.roesch@env.ethz.ch)

**Abstract.** In this study, we describe the design and testing of a high output stability, constant liquid feed nebulizer using the Venturi principle to generate liquid particles from solutions. This atomizer, the PRinted drOpleT Generator (PROTeGE) was manufactured using stereolithography (SLA) printing. Different concentrations of ammonium sulfate solutions were used to characterize the size and number concentration of the generated particles. A comparison of a 3D printed 0.5 mm orifice against a commercially available 0.5 mm brass orifice using the same ammonium sulfate solution was also performed. The particle number concentration generated with the printed orifice was higher, by ~2x, than the particle number concentration generated with the brass orifice.

PROTeGE is also capable of dispersing polystyrene latex spheres (PSLs) for calibration purposes. The particle number concentrations obtained in this study ranged from ~10,000 cm$^{-3}$ for 0.75 μm to ~100 cm$^{-3}$ for 5.0 μm PSL particles with a dependence on the concentration of the dispersed solution. For the different concentrated ammonium sulfate solutions particle

number concentrations from ~14,000 cm$^{-3}$ for 0.1 g L$^{-1}$ to 7,600 cm$^{-3}$ for 5.0 g L$^{-1}$ were
measured. An additional measurement with a Scanning Electrical Mobility System (SEMS)
was performed for the 0.6 g L$^{-1}$ solution to measure particles in the size range of 10 nm to 1000
nm. The generated particle number size distributions showed a maximum at 50 nm with particle
number concentrations of ~40,000 cm$^{-3}$. PROTeGE is easy to manufacture and operate, low in
maintenance, and cost-effective for laboratory and field generation of particles from aqueous
media in a size range of 10 nm to 5000 nm.

**1 Introduction**

Reliable and cost-effective particle generation methods are necessary for applications
where a well-defined mono- or polydisperse particle concentrations and size is required. High
concentrations of monodisperse aerosol particles (>10$^6$ particles cm$^{-3}$) in the nm to μm diameter
size range can be created by instruments utilizing vapor condensation and electrospray
techniques. Similar concentrations and size ranges of aerosol particles can be produced from
bulk solutions using a variety of instruments such as vibrating-orifice aerosol generators and
ultrasonic nebulizers. Another common technique used to generate liquid aerosol particles is
by pressurized-air nebulization. In this method, compressed air is utilized to shatter a solution
into small aerosol droplets with a specific size distribution (Swiderska-Kowalczyk et al., 1997).
Pressurized-air nebulizers have been used in numerous studies which require the generation of
aqueous aerosol particles. For example, Wang et al. (2019) used a nebulizer to create
ammonium sulfate seed particles in cloud droplet activation studies. Kong et al. (2018) created
aqueous aerosol particles used to study deliquescence and ice nucleation in sea salt particles.
These are two of many studies focused on the influence of aerosol particles in the Earth's
atmosphere. Pressurized-air nebulizers are also used in pharmaceutical applications to produce
nanometer-sized drug particles with a specific size distribution (Eerikainen et al., 2003). It is
clear that the pressurized-air nebulizer is a ubiquitous instrument for studies or applications
requiring aqueous aerosol particle generation. Although effective, commercial particle
generation instruments may not be economically feasible for all research and teaching
institutions wishing to perform these types of experiments.

57        Advances in 3D printing have made it possible to rapidly fabricate high-resolution (μm-

scale) devices. Stereolithography (SLA), a form of 3D printing technology, creates objects in
layers through the use of photopolymerization. In conjunction with Computer Aided Design
(CAD), or Computer Aided Manufacturing (CAM) software, an ultraviolet (UV) laser is used
to trace a pre-programmed design on to the surface of a photopolymer contained in a vat. The
resin is photochemically solidified and forms a single layer of the desired object. The Form 2
SLA 3D printer (Formlabs, Inc.), used in this work, is capable of creating objects with a layer
thickness of 25 μm.

65        The Form 2 was previously used to fabricate PRIZE, a compact fluidized bed aerosol

generator (Roesch et al. 2017). It was found that PRIZE was able to successfully disperse
aerosol particles from dry material without creating artifact particles (particles generated from
the material used to fabricate the generator). The impetus for this study, similar to the study
presented in Roesch et al. 2017, was to fabricate a low-cost, constant pressure nebulizer, using
a SLA 3D printer, PROTeGE. In the following two sections we describe how PROTeGE is
designed and manufactured. The experimental setup and performance tests using different PSL
and ammonium sulfate solutions is discussed. Three types of experiments were conducted to
demonstrate the performance of PROTeGE: (1) an aerosol production experiment using four
different sizes of PSL's, (Polysciences Inc., NIST traceable) ranging from 0.75 to 5.0 μm, (2)
experiments where different concentrations of ammonium sulfate solutions were dispersed and
monitored over time with an optical particle sizer (OPS, Model 3330, TSI Inc.) and for the 0.6
g $L^{-1}$ also with a Scanning Electrical Mobility Spectrometer (SEMS, BMI Inc.), (3) an
experiment comparing the performance of a printed 0.5 mm orifice to a 0.5 mm commercial
brass orifice using the same ammonium sulfate solution of 0.6 g L$^{-1}$.


**2 Methods**
**2.1. Design**
PROTeGE was designed using a computer aided design (CAD) program (Solidworks 2015,
Dassault Systems). There are two versions: the first is printed as a single part including a 0.5
mm diameter orifice (Fig. 1), while the second, featuring the same inner and outer dimensions,
has an exchangeable nozzle to use various machined orifice diameters. The 0.5mm (0.02 in)
orifice used in the comparison is a commercially available brass nozzle (Part Number
2943T887, McMaster-Carr) that is threaded into the pressurized air inlet of PROTeGE (Fig.
1d). This modular feature enables rapid exchange of nozzles with a different orifice diameter
using the same printed unit. In contrast, the exclusively printed version of PROTeGE has a
fixed orifice diameter for continuous and simple operation. Both versions are based on the
generator designs of May, 1973 and Liu & Lee, 1975. Unlike the repurposing of existing
nebulizers built for other uses, such as medical applications (Reisner et al., 2001), PROTeGE
was designed specifically for research applications e.g. instrument calibration and particle
generation, similar to other custom-built nebulizers (Wex et al., 2015). In contrast to other
generation systems, which are typically machined, PROTeGE is made from photopolymer resin
(e.g. FLGPCL02, Formlabs Inc.), weights only ~50 g, and can therefore feature smaller overall
dimensions.
The front inlet to PROTeGE is a 6.35 mm (0.25 in) barbed tube to connect to a pressurized
airflow. The inlet is end-capped by the orifice. Directly following, and perpendicular oriented
to the orifice, is the liquid feed 3.18 mm (0.125 in) inlet to the dispersible solution. Excess
liquid from the nebulizing process exit the chamber through a 6.35 mm outlet at the bottom,
dripping directly back into the feed bottle. Aqueous particles exit the chamber through the 9.53
mm (0.375 in) aerosol outlet at the top. The overall dimensions of PROTeGE are 17 x 45 x 65
mm (width, depth, height). Designed CAD files were converted to style files (.stl) to be readable
by the 3D printer software (PreForm, Formlabs Inc.).

## 2.2. Manufacturing

The manufacturing and post-processing of the parts was performed as described by Roesch et
al., 2017 using the same 3D printer software, clear photopolymer resin (FLGPCL02, Formlabs
Inc.) and 3D SLA printer (Form 2, Formlabs Inc.). Modifications were made to the dimensions
of the default contact points of the printing scaffolding; in this study scaffolding was reduced
to 0.45 mm due to the overall smaller geometry of PROTeGE (i.e., lower mass needing to be
supported). Using a resolution of 100 μm, eight complete PROTeGEs can be printed on the
build surface at the same time, taking ~8 h.
A custom UV box was used to post-cure the printed parts. Inside the box, the printed parts were
placed on a slow moving turntable to be illuminated equally from all sides by 28 high-power
LEDs emitting at 405 nm. It should be noted that the curing time depends on the size and wall
thickness of the printed part; one hour per mm wall thickness is suggested. The post-curing
time for PROTeGE was ~1 h. For more detailed information see the manufacturing section in
Roesch et al., 2017. The cost to produce one PROTeGE is around ~\$2.50 depending on the type
of resin and the percentage of scaffolding used. The commercial brass nozzle costs <\$10, so a
total PROTeGE costs under \$15. For users with no access to a 3D printer, it is also possible to
upload the .stl file for PROTeGE (provided at https://www.thingiverse.com/thing:4444498) to
an online print service. Pictures of PROTeGE and post processing details for the instrument are
also provided on the data repository.

## 2.3. Experimental setup

A schematic of the experimental setup including the relevant flow rates used in this study is shown in Fig. 2. Dry, filtered, pressurized air was used as the carrier gas. The input flow rate of 1.7 L min$^{-1}$ (at 35 psi) into PROTeGE was controlled by a rotameter (MR3A, Omega Engineering). A high velocity jet is created by the expansion of the pressurized air through the orifice. As a result, the pressure behind the orifice drops, the liquid is pulled upward from the feed bottle, and the high velocity jet disperses the liquid solution into droplets. Large droplets that are unable to follow the streamlines through the aerosol outlet are removed by impaction at the curved wall; these drip back as excess liquid into the feed bottle through a drain outlet at the bottom of PROTeGE. For these experiments, the droplets from the aerosol outlet were subsequently dried using a silica gel drier. Downstream, the flow of residuals (remaining solid cores of the droplets) was split into two flows. The first is sent to the OPS, to determine PNSDs in the size range of 0.3 to 10 µm and the remainder through a filter (IDN-4G, Parker) open to lab. Unless otherwise noted, all experiments presented here were performed using the brass nozzle with a 0.3 mm orifice and the previously stated pressures and flow rates.

## 3. Results

In this study three types of experiments were conducted: (1) an aerosol production experiment using four different sizes of PSL's, ranging from 0.75 to 5.0 µm, (2) an application experiment where different concentrated ammonium sulfate solutions were dispersed and monitored over time with the OPS and for the 0.6 g L$^{-1}$ also with the SEMS, (3) a comparison experiment on the performance of a printed 0.5 mm orifice versus a 0.5 mm commercial brass orifice using the same ammonium sulfate solution. For all experiments the PNSDs of the droplet residuals (i.e., after drying) were measured with the OPS.

Prior to each experiment, PROTeGE was immersed in a jar with Destilled De-ionized (DDI) 18.2 MΩ•cm Millipore water and sonicated for 10 minutes in an ultrasonic bath to ensure clean inner surfaces. Afterwards, PROTeGE was dried using pressurized nitrogen and connected to

the setup. Each of the four samples was prepared in a separate 100 ml glass bottle, using 80 ml
of DDI water plus multiple drops of the respective PSL solution (0.75 µm, 1.5 µm, 2.0 µm, 5.0
µm). The generated number concentration of PSL particles strongly depends on the
concentration of the prepared PSL sample. Therefore, the higher the concentration of the
solution, the higher the generated particle number concentration. A time-series measurement of
420 seconds was performed for each of the four PSL samples. The obtained PNSDs showed
particle number concentrations of ~10,000 $cm^{-3}$ for 0.75 µm PSL particles, ~1,000 $cm^{-3}$ for 1.5
µm PSL particles, ~800 $cm^{-3}$ for 2.0 µm PSL particles to ~100 $cm^{-3}$ for 5.0 µm PSL particles
(Fig. 3). All four investigated PSL samples showed a narrow PNSD except the 5.0 µm PSL
sample where a fraction of sub-micrometer particles was detected (Fig. 3d). This fraction of
particles likely originates from the solution matrix in which the PSLs are suspended. Overall
the generated PNSDs were stable over their measured period of time while only the 1.5 µm
PSL sample showed a slight decrease. These data show that the curved design of the chamber
enables PROTeGE to disperse PSL particles with diameter up to 5.0 micrometer.
In addition to the PSL measurements, ammonium sulfate experiments with different solution
concentrations (0.1 g $L^{-1}$, 0.6 g $L^{-1}$, 5.0 g $L^{-1}$) and an experiment to determine the performance
of the printed 0.5 mm orifice versus the 0.5 mm brass orifice using the same ammonium sulfate
solution of 0.6 g $L^{-1}$ were conducted. The three solutions were again prepared in separate 100
ml glass bottles using DDI 18.2 MΩ•cm Millipore water. The cleaning procedure of PROTeGE
was identical to the one described above for the PSL experiments.
For the lowest concentration of aqueous ammonium sulfate solution (0.1 g $L^{-1}$) the maximum
particle number concentration of ~14,000 $cm^{-3}$ was observed in the 0.3 µm bin of the OPS (Fig.
4a). The overall width of the generated PNSDs ranged from 0.3 µm up to 0.6 µm. Compared to
the higher concentrated solutions of 0.6 g $L^{-1}$ and 5.0 g $L^{-1}$ this is rather narrow. Dispersing the
0.6 g $L^{-1}$ solution of ammonium sulfate generated PNSDs in the range of 0.3 µm up to 1.5 µm
with a maximum particle number concentration of ~8,200 $cm^{-3}$ in the 0.5 µm detection bin of
the OPS (Fig. 4b). For the highest concentration of 5.0 g $L^{-1}$ the generated PNSDs ranged from
1.0 µm to 2.4 µm with a maximum particle number concentration of ~7,600 $cm^{-3}$ (Fig. 4c).
For the 0.6 g $L^{-1}$ solution an additional experiment using the SEMS instrument was performed.
The size range was scanned from 10 nm to 1000 nm with a resolution of 60 bins and a sampling
rate of 1 second per bin. The maximum particle number concentration was found at ~50 nm
with ~40,000 $cm^{-3}$. The average PNSD for a 420 second sampling period is shown in Fig. 4d.
During the experiment the generated size distributions did not change over time. Combining
the obtained size distributions from SEMS and OPS shows that PROTeGE is capable of
generating particles as small as 10 nm up to 2.4 µm based on the dispersed ammonium sulfate
solution.
Overall, the generated particle number concentrations for the different tested ammonium sulfate
solutions are high enough (> 1,000 $cm^{-3}$) to operate particle size selection instruments
downstream of PROTeGE, e.g. a differential mobility analyzer (DMA), assuming ~10% of the
introduced particles are selected as monodisperse aerosol particles.
In order to investigate the performance of an integrally printed 0.5 mm orifice versus the
commercial brass nozzle the same 0.6 g $L^{-1}$ ammonium sulfate solution was used. Visual
observations of the printed 0.5 mm orifice showed that there were sometimes imperfections and
asymmetries in the roundness of the printed orifice. It was therefore necessary to post-drill the
printed orifice with a 0.5 mm bit by hand, which was done for the PROTeGE used in experiment
described here. The generated particle number concentration with the printed orifice was
~4,500 $cm^{-3}$ versus ~2,400 $cm^{-3}$ for the brass nozzle with minimal difference in PNSD shape. The
printed nozzle exhibited a broader shoulder from 0.3 µm to 0.5 µm with higher particle number
concentrations than the brass nozzle. The latter showed a decline with the highest particle
number concentration at 0.3 µm leveling out to 1.0 µm. Overall, the width of both PNSDs
ranged from 0.3 µm to 1.0 µm (Fig. 5).

## 4. Conclusion

In this study, we described the design and the performance of a low-cost particle generator – PROTeGE. The experiments presented here show the PROTeGE capability for generating four different sizes of PSL particles between 0.75 and 5.0 µm with resulting particle number concentrations between 100 cm$^{-3}$ and ~10,000 cm$^{-3}$. This enables PROTeGE to be used as a simple and cost effective particle generation unit for calibration purposes. In addition, we show the results of experiments using different concentrations of ammonium sulfate solutions. The generated PNSDs ranged from 10 nm up to 2.4 µm, based on the dispersed solution, with stable output concentrations demonstrating that particle size selection instruments can be used downstream of PROTeGE. Finally, we compared the performance of a commercially machined brass 0.5 mm orifice to a 3D printed 0.5 mm orifice. Both nozzles showed PNSDs ranging from 0.3 µm to 1.0 µm with a broader shoulder for the printed nozzle between 0.3 µm and 0.5µm compared to the brass orifice. Also the total particle number concentration generated with the printed orifice was almost two times higher than the total particle number concentration generated with the brass orifice.

Due to the low cost of PROTeGE (~$15, material costs) multiple generators can be used in parallel to reduce experimental time while running more samples.

Data availability: The .stl files for PROTeGE are available at https://www.thingiverse.com/thing:4444498.

Authors contribution: MR and DJC contributed both equally to the manuscript. The experiments were conducted by MR.

Competing interests: The authors declare that they have no conflict of interest.

234

235

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

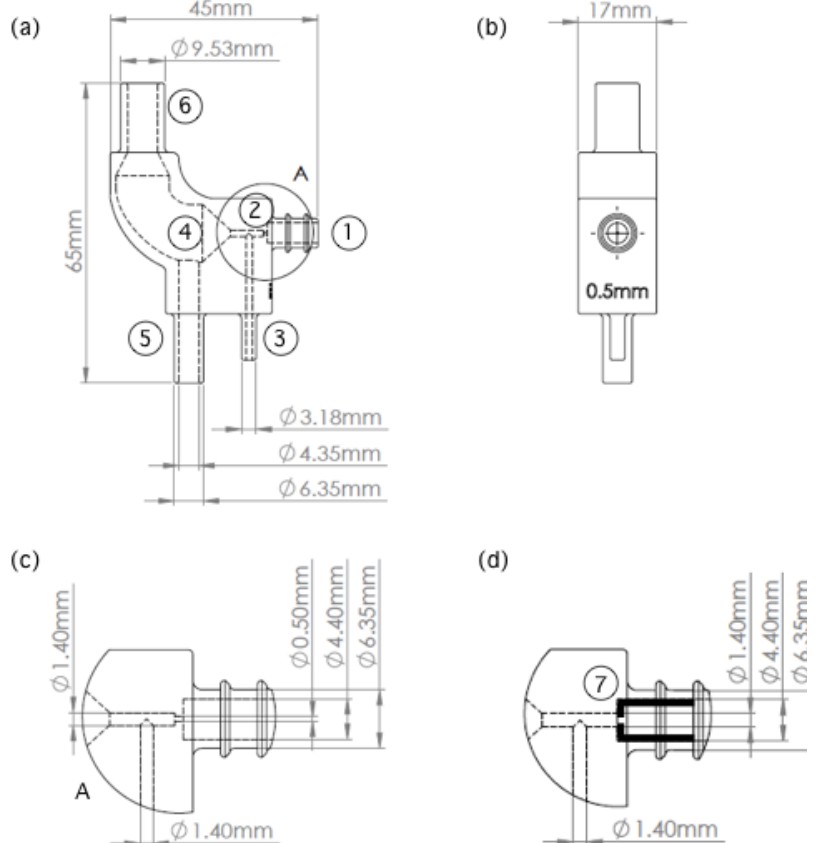


**Figure 1:** Dimensioned drawings of PROTeGE: (a) side view; (b) front view; (c) detailed view of the inlet section with the printed 0.5mm orifice; (d) detailed view of the inlet section with the exchangeable nozzle/orifice. PROTeGE consists of: (1) a pressurized air inlet, (2) a printed 0.5mm orifice, (3) a liquid feed inlet, (4) a central impaction chamber, (5) the drain outlet, (6) an aerosol outlet, and an optional (7) exchangeable orifice.


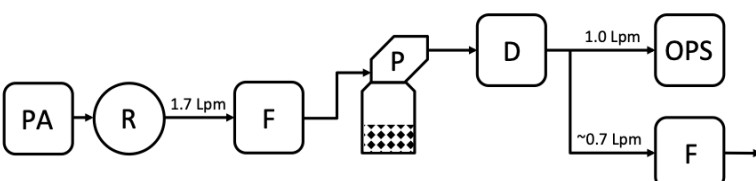


**Figure 2:** Schematic of the experimental setup used in this study. A dry pressurized air flow (PA) was passed through a rotameter (R) to control the flow rate and a filter (F) upstream of PROTeGE (P). Generated droplets were dried with silica gel (D) before the flow of particles was directed into an OPS with excess flow discarded through a filter (F).

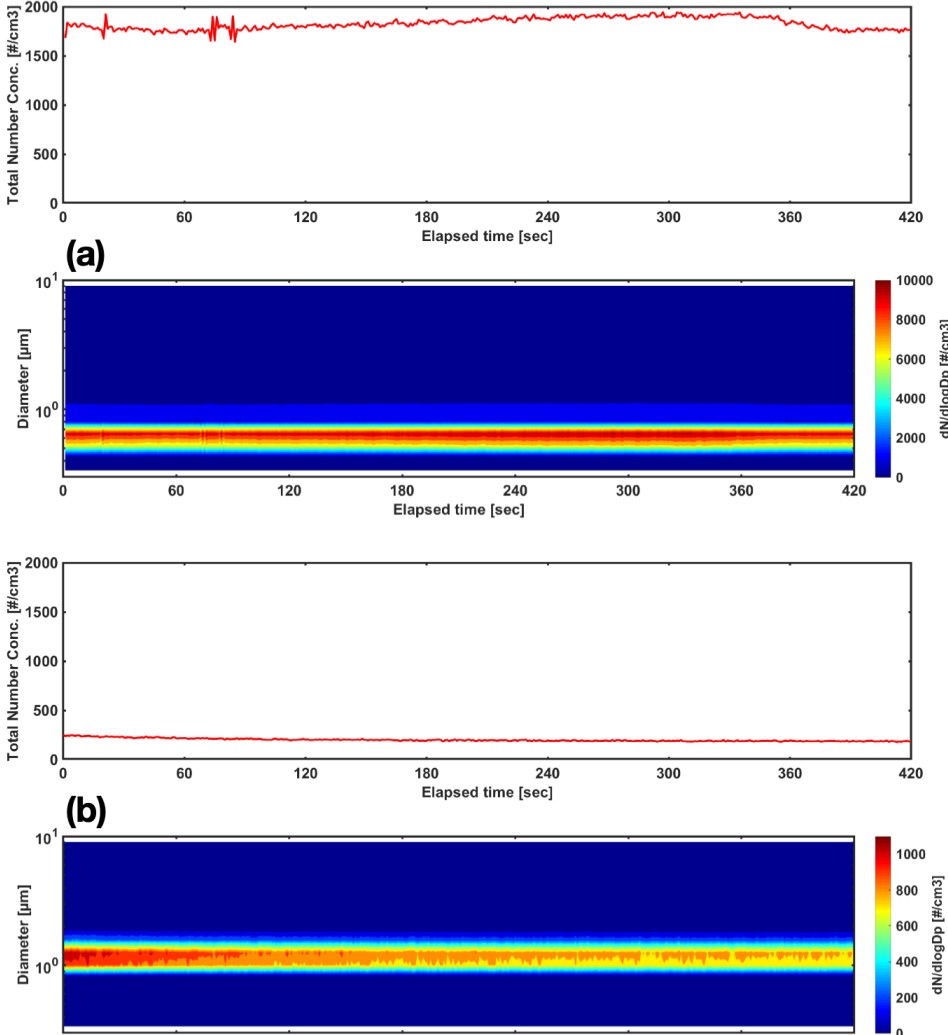








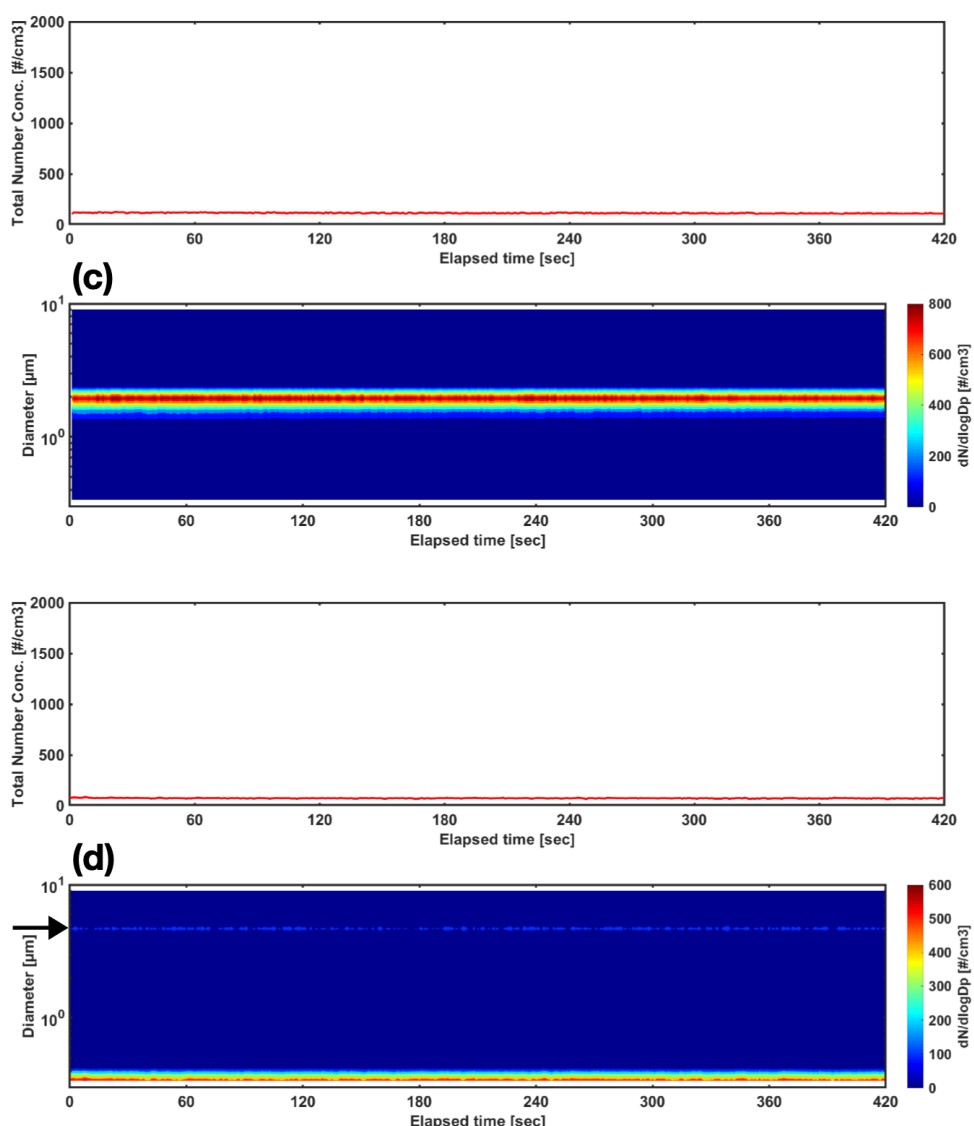




**Figure 3:** Total particle number concentrations and particle number size distributions as function of time for different sized polystyrene latex spheres generated with PROTeGE and detected by the OPS: (**a**) 0.75 µm; (**b**) 1.5 µm; (**c**) 2.0 µm; (**d**) 5.0 µm where the black arrow denotes the PSL size and the small particles represent atomized matrix material (see text for details).



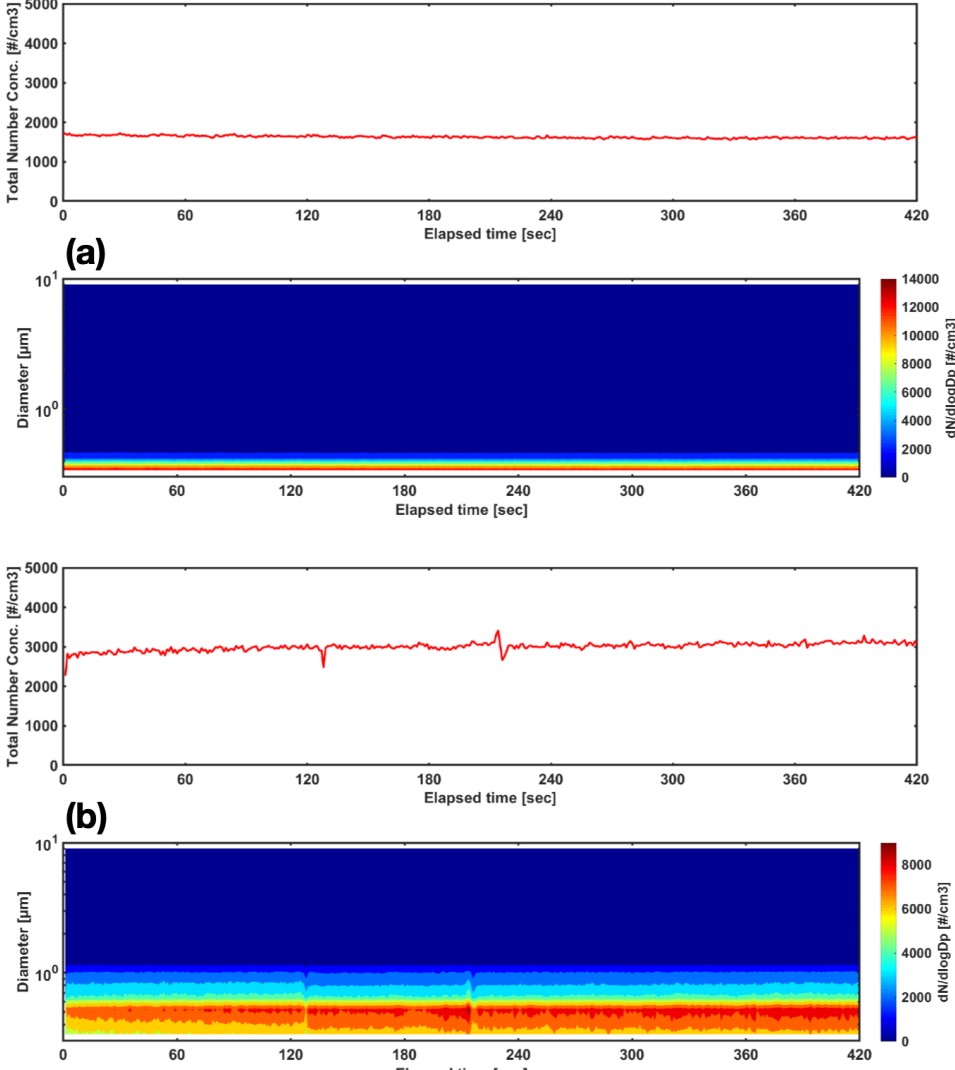


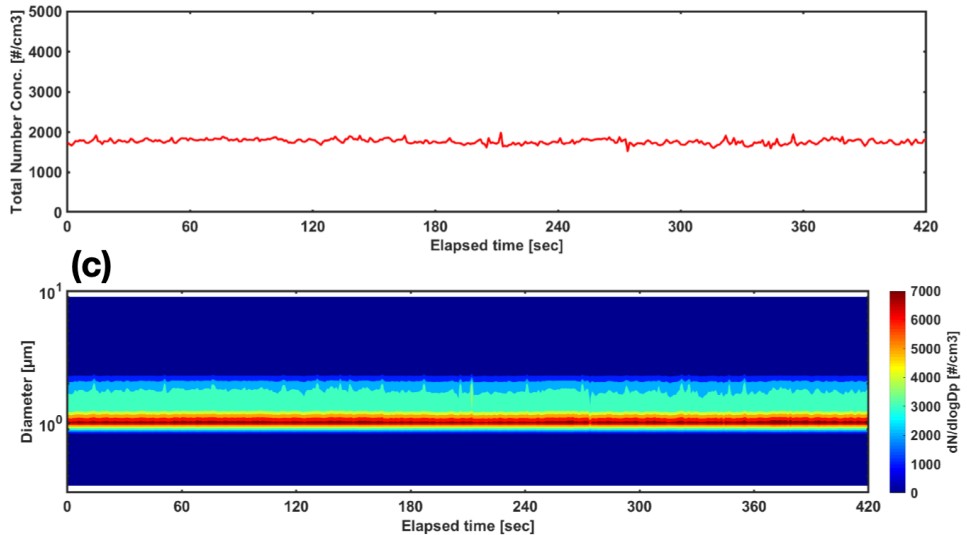

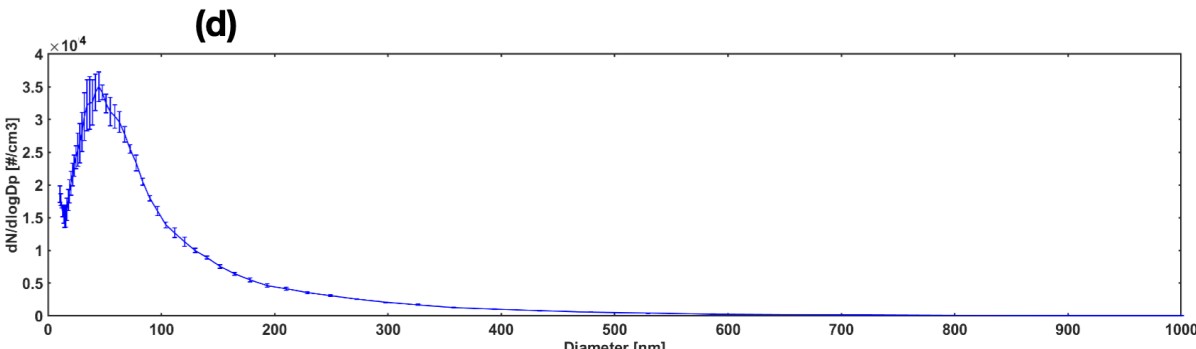

**Figure 4:** Total particle number concentrations and particle number size distributions as a function of time of aqueous ammonium sulfate particles: **(a)** 0.1 g L$^{-1}$; **(b)** 0.6 g L$^{-1}$; **(c)** 5.0 g L$^{-1}$; **(d)** Average particle number size distribution from 10 nm to 1000 nm measured with the SEMS for aqueous ammonium sulfate particles of 0.6 g L$^{-1}$.


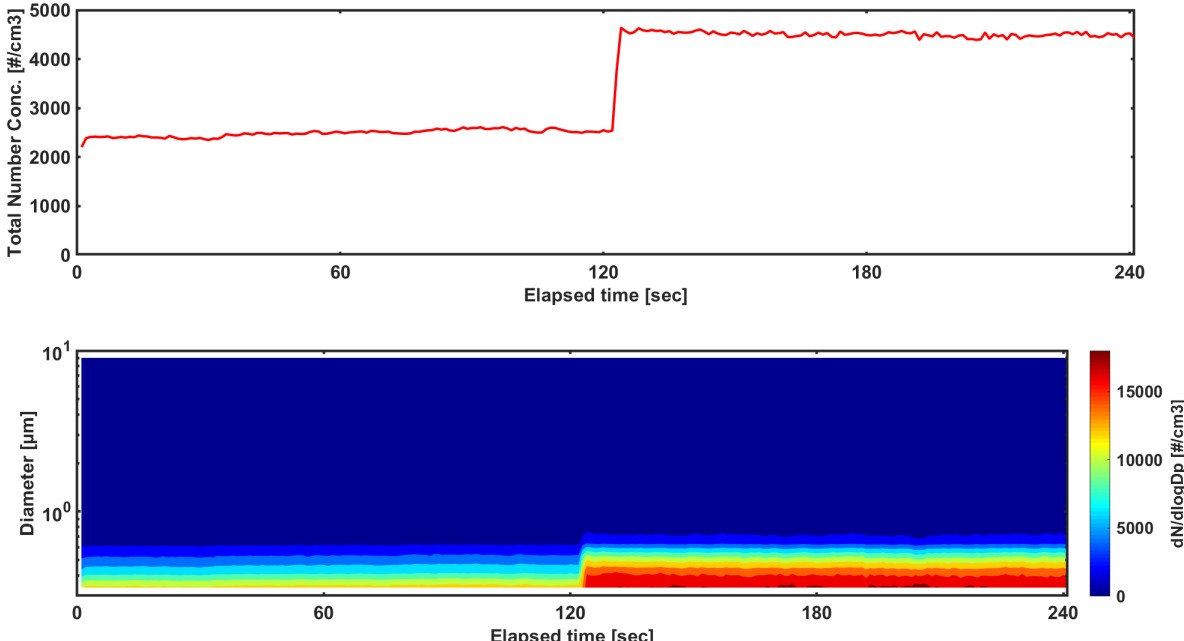







**Figure 5:** Total particle number concentrations and particle number size distributions as a function of time of aqueous
ammonium sulfate particles at 0.6 g L$^{-1}$: 0.5mm brass orifice, from 0-120 sec;  0.5mm printed orifice, from 120-240 sec.