# Peer review of "Aqueous particle generation with a 3D printed nebulizer"

_Atmospheric Measurement Techniques, 2020_

## Referee Comment (RC1) · Anonymous Referee #2 · 23 Jun 2020

This is a rather short manuscript on the 3D printing of a nebulizer. The novelty might justify publication but the manuscript clearly needs to be improved to be more rigorous and discuss more critically even the data shown.

Issues:

The manuscript keeps insinuating a "low cost" nebulizer (e.g. L44).. but does not provide any cost reference neither for the competition nor for the actual set-up. In terms of competition, depending on the particle range of the aerosols that are actually being generated, medical nebulizers are «100$ with the actual disposable nebulizer being ∼3$. Your 3D printed ones are not that cheap as they 1) require a 5K printer 2) the raw plastic material and 3)in your set-up a brass nozzle at ∼10$.. Therefore it would be critical to have a more nuanced discussion. Also the commercial (expensive)

[Figure]

devices might have a much better performance (see next point).

The data analysis of the particles generated is very superficial. Only focus is on number concentrations generated (again context of commercial and alternative systems is inexistent). There is no significant discussion in the manuscript on the actual distributions generated and their stability. Number concentrations is one aspect but what about distributions. Also the "heatmaps" provided the distributions are very confined to a small area and no effort was made to quantitatively analyze that data… e.g. how does the mode of the distribution changes over time.. or does not? What is the broadness of the distribution? any quantitative distribution metric and how this relates to commercial systems or applications. For the very least the discussion on figure 5 needs to be extended… Saying that they are similar is not true, there is a lot of difference (y axis is log) and it does matter.

Related the PSL "calibrations" seem disconnected to what can be achieved with the salt solutions?

All experiments except the comparison were done with the brass nozzle? This id stated a little bit as an aside given that the whole paper makes it sound as it whole nebulizer was 3D printed when in fact the most critical part (nozzle) was not.. but it was brass and purchased. One wonders why if the conclusion of figure 5 is that they are equivalent? Could you comment on this?

Other issues

Abstract: would be more informative to actually say what that the comparison with the brass orifice revealed rather than just say it was done. Results should be summarized in the abstract not just written what was done.

On the other hand the typical last paragraph of the introduction, where one typically says that will eb discussed in the manuscript is missing…

Then again the first paragraph of the results (L112-115) is actually just that: saying

what will be done... and this would belong as last paragraph of the intro. This is just an odd way of writing a manuscript.

L21 Please use comma for thousands to ease reading

Please be precise on the brass nozzle and diameter used. The cat number for McMaster-Carr shows orifices in inch (of various sizes).... Does the 0.5mm mean you used the 0.02 inch one?

L122 use center dot as multiplication sign not a star

L120: please explain double distilled deionized.? Millipore systems do not distill? Where does tht DDI come from?

L149 "will be sufficient enough" can you be more quantitative.. what do you consider sufficient?

Figure 1: what is the rationale behind the numbering.. why does (2) jump to pane d....

Figure 4 there is no discussion at all why the time scale varies so much between panels a,b and c... Please discuss in the text what you want to show going from a) 1800sec to b) 450 sec to c) 20000 sec

Figure 4 and 5. I suggest that the top panels with number concentrations should present the same extent of range.... To have a visual meaning.

Right now the resolution is so poor in the figures that the legend of the top panels (fig 4 and 5) are hardly readable.
* * *

---

## Referee Comment (RC2) · Anonymous Referee #3 · 11 Aug 2020

Rösch and Cziczo present characterization of a 3D printed nebulizer for aerosol generation. The instrument is tested by nebulizng PSL spheres and ammonium sulfate and measuring the particle size distribution using an optical particle spectrometer between 300 nm and 10 micron. Heatmaps of particle size distribution between 160s and 3x10^4 s are presented, which are intended to show the stability of the atomizer.

Overall, this is an interesting contribution. The paper is short and easy to digest. The paper might be publishable in AMT if the authors are more forthcoming about the details of the instrument. However, to make this work publishable either more experiments are needed that demonstrate that the technique is an improvement over existing technologies (or at least not a regression), or detailed open access publication of the plans is needed to increase accessibility of the technique. Either require major revisions to

the manuscript.

Specific comments

Particle generation by nebulization is a very well established, commercially available, and widely used technique. The experiments amount to a couple of days worth of work in an aerosol laboratory outfitted with basic equipment. A new way to manufacture such an item in itself is in my opinion insufficient to warrant publication. To this referee, the criteria for publication are either a significant improvement over existing technology or an increase in accessibility of the technique.

Is the technique improved over existing technology? The authors do not show data for D < 300 nm, the authors do not quantify the total number concentration of drops or typical droplet size produced, the authors do not quantify the composition of particles produced, the range of solvents that can be used (I guess some organic solvents might be problematic), the degree to which mixed particles (e.g. ammonium sulfate + organic compounds and preserving the ratio in the atomized particles) can be generated from aqueous stock solutions, or the minimum aerosol diameter that can be generated, which is determined by cleanliness of the solvent and drop size, the maximum time the instrument can run unattended, the degree of drying that is needed, and the range of pressure and flow rates at which the atomizer produces particles. All of these are critical to evaluate if such a device is suitable for application in laboratory research, including for instrument calibration. Thus, the answer to the question is no.

Experiments including an SMPS to measure the full size distribution should be included. Experiments should systematically characterize the output for a much wider range of inputs (solvent, composition, solute weight percent) and analyze the results to infer drop number size and concentration. Ideally composition measurements of mixed particles are included to test for artifacts such as dissolution of the plastic and faithful representation of stock solution (e.g. adsorption of organics while the liquid passes through the atomizer).

Does the work increase accessibility of the technology? The paper states that the authors were able to build this device, which is nice. However, there is no benefit to the community if it is not widely shared on how to do that. The authors state that STL files are "available upon request". This is insufficient. In my experience, share requests are often conveniently ignored or come with strings attached by the sharer. They present an unnecessary barrier. Thus, the answer to the question is no.

If the authors want this instrument to be a low cost, self-manufacture replacement, the authors should provide the STL files as a supplement or make them available in an archived repository. The paper should include an itemized list what people need to purchase, including part numbers and cost estimates. A photo of the instrument would be a good addition to the paper. The printing could be performed by a 3D printing service and ordered with a couple of clicks. Quotes can be generated from online vendors within minutes (e.g. sculpteo) by uploading the STL file. Assembly instruction should be provided. Comments about alternative print materials should be made and the precision that is needed for printing (is 100 micron the limit?). All of the designed parts should be made available using open licenses, e.g. the CERN open hardware license (https://www.ohwr.org/cernohl). Such a device would be very welcome and provide a platform where anyone could build, try, and characterize the output for themselves. In this case, the likely performance limitations and/or deficits in characterization raised earlier are less critical.

Irrespective the route the authors wish to pursue, the authors need to comment on the technical limitations above in the revised paper. The authors should also compare cost and performance to other techniques. For example, the TSI atomizer is ∼\$3k and very stable, and very well characterized. Small medical nebulizers (pressure and ultrasonic) can be obtained for < \$30 and are more than sufficient to generate good aerosol for shorter duration (5-15 min). It might be useful to juxtapose data from these side-by-side and discuss use cases for the printed design.

---

## Author Comment (AC1) · 15 Oct 2020

Authors' response

Dear Referees,

Thank you for your comments to improve the manuscript.
We have addressed your points and made the following changes to the
manuscript.
We believe that the manuscript is greatly improved and is now ready for
publication in AMT.

Thank you.

Best regards,

Michael Rösch and Dan Cziczo

**Referee 2:**

*„The manuscript keeps insinuating a "low cost" nebulizer (e.g. L44).. but does not provide any cost reference neither for the competition nor for the actual set-up. In terms of competition, depending on the particle range of the aerosols that are actually being generated, medical nebulizers are «100$ with the actual disposable nebulizer being 3$. Your 3D printed ones are not that cheap as they 1) require a 5K printer 2) the raw plastic material and 3)in your set-up a brass nozzle at 10$.. Therefore it would be critical to have a more nuanced discussion. Also the commercial (expensive) devices might have a much better performance (see next point)."*

We added the material price to print one PROTeGE plus the price for the brass nozzle to the manuscript for reference. In Addition, there are online print services that can 3D print parts and most universities now have fabrication labs or in-house 3D printing capabilities.

„The cost to produce one PROTeGE is around ~$2.50 depending on the type of resin and the percentage of scaffolding used. The commercial brass nozzle costs <$10, for a total PROTeGE cost under $15. For users with no access to a 3D printer, it is also possible to upload the .stl file for PROTeGE (provided at no cost at https://www.thingiverse.com/thing:4444498) to an online print service. Pictures of

PROTeGE and post processing details for the instrument are also provided on the data repository."

*„The data analysis of the particles generated is very superficial. Only focus is on number concentrations generated (again context of commercial and alternative systems is inexistent). There is no significant discussion in the manuscript on the actual distributions generated and their stability. Number concentrations is one aspect but what about distributions. Also the "heatmaps" provided the distributions are very confined to a small area and no effort was made to quantitatively analyze that data… e.g. how does the mode of the distribution changes over time.. or does not? What is the broadness of the distribution? any quantitative distribution metric and how this relates to commercial systems or applications."*

Thank you for your detailed comment.
Both aspects particle number concentrations and the corresponding size distributions are described in the manuscript through the heat maps. One could also derive the broadness of the size distribution and how it changes over time from there. We performed an additional experiment using a SEMS instrument to detect the generated PNSD in the size range of 10 nm to 1000 nm. A comparison to other commercial systems is beyond the scope of this study as we are limited by instrumentation.
We changed the paragraph in the manuscript as follows:

„A time-series measurement of 420 seconds was performed for each of the four PSL samples. The obtained PNSDs showed particle number concentrations of ~10,000 cm$^{-3}$ for 0.75 $\mu$m PSL particles, ~1000 cm$^{-3}$ for 1.5 $\mu$m PSL particles, ~800 cm$^{-3}$ for 2.0 $\mu$m PSL particles to ~100 cm$^{-3}$ for 5.0 $\mu$m PSL particles (Fig. 3). All four investigated PSL samples showed a narrow PNSD except the 5.0 $\mu$m PSL sample where a fraction of sub-micrometer particles was detected (Fig. 3d). This fraction of particles likely originates from the solution matrix in which the PSLs are suspended. Overall the generated PNSDs were stable over their measured period of time while only the 1.5 $\mu$m PSL sample showed a slight decrease. These data show that the curved design of the chamber enables PROTeGE to disperse PSL particles with diameter up to 5.0 micrometers."

*„Related the PSL "calibrations" seem disconnected to what can be achieved with the salt solutions?"*

The PSL measurements/calibrations are essential for most end users. Our measurements show that PROTeGE can be used for multiple types of particles including the generation of PSL particles.

*„All experiments except the comparison were done with the brass nozzle? This id stated a little bit as an aside given that the whole paper makes it sound as it whole nebulizer was 3D printed when in fact the most critical part (nozzle) was not.. but it was brass and purchased. One wonders why if the conclusion of figure 5 is that they are equivalent? Could you comment on this?"*

Both nozzle types generate the same width of the size distribution with the only difference in their particle number concentration for certain bin sizes.
Please see the changes we made in the manuscript based on your next comment.

*„For the very least the discussion on figure 5 needs to be extended…Saying that they are similar is not true, there is a lot of difference (y axis is log) and it does matter."*

We extended the discussion on Figure 5 and added the following paragraph to the manuscript:

"The generated total particle number concentration with the printed orifice was ~4,500 cm$^{-3}$ versus ~2,400 cm$^{-3}$ for the brass nozzle with minimal difference in PNSDs shape. The printed nozzle exhibited a broader shoulder from 0.3 µm to 0.5 µm with the higher particle number concentrations than the brass nozzle. The latter showed a decline with the highest particle number concentration at 0.3 µm leveling out to 1.0 µm. Overall, the width of both PNSDs ranged from 0.3 µm to 1.0 µm (Fig. 5)."

*„Abstract: would be more informative to actually say what that the comparison with the brass orifice revealed rather than just say it was done. Results should be summarized in the abstract not just written what was done."*

We added the following paragraphs to the abstract:

"A comparison of a 3D printed 0.5 mm orifice against a commercially available 0.5 mm brass orifice using the same ammonium sulfate solution was also performed. The particle number concentration generated with the printed orifice was higher, by ~2x, than the particle number concentration generated with the brass orifice."

"For the different concentrated ammonium sulfate solutions particle number concentrations from ~14,000 cm-3 for 0.1 g L$^{-1}$ to 7,600 cm-3 for 5.0 g L$^{-1}$ were measured."

**„On the other hand the typical last paragraph of the introduction, where one typically says that will eb discussed in the manuscript is missing…Then again the first paragraph of the results (L112-115) is actually just that: saying what will be done…and this would belong as last paragraph of the intro. This is just an odd way of writing a manuscript."**

We added a new paragraph to the end of the introduction:

"In the following two sections we describe how PROTeGE is designed and manufactured. The experimental setup and performance tests using different PSL and ammonium sulfate solutions is discussed. Three types of experiments were conducted to demonstrate the performance of PROTeGE: (1) an aerosol

production experiment using four different sizes of PSL's, (Polysciences Inc., NIST traceable) ranging from 0.75 to 5.0 μm, (2) experiments where different concentrations of ammonium sulfate solutions were dispersed and monitored over time with an optical particle sizer (OPS, Model 3330, TSI Inc.) and for the 0.6 g L$^{-1}$ also with a Scanning Electrical Mobility Spectrometer (SEMS, BMI Inc.), (3) an experiment comparing the performance of a printed 0.5 mm orifice to a 0.5 mm commercial brass orifice using the same ammonium sulfate solution of 0.6 g L$^{-1}$."

**„L21 Please use comma for thousands to ease reading"**

We changed all notations to comma for thousands for ease reading throughout the manuscript.

**„Please be precise on the brass nozzle and diameter used. The cat number for McMaster-Carr shows orifices in inch (of various sizes)… Does the 0.5mm mean you used the 0.02 inch one?"**

We provide information on the orifice in the manuscript and also on the part number to select the correct one as follows:

„The 0.5 mm orifice used in the comparison is a commercially available brass nozzle (Part Number 2943T887, McMaster-Carr) that is threaded into the pressurized air inlet of PROTeGE (Fig. 1d)."

***„L122 use center dot as multiplication sign not a star"***

We changed the multiplication sign to center dot throughout the manuscript.

***„L120: please explain double distilled deionized.? Millipore systems do not distill? Where does tht DDI come from?"***

Thank you for the indication. We corrected the term DDI in the manuscript according to the manufacturer as follows:

„Destilled De-ionized (DDI)"

***„L149 "will be sufficient enough" can you be more quantitative.. what do you consider sufficient?"***

We changed Line 149 to a more quantitative statement as follows:

„ Overall, the generated particle number concentrations for the different tested ammonium sulfate solutions are high enough (> 1,000 $cm^{-3}$) to operate particle size selection instruments downstream of PROTeGE, e.g. a differential mobility analyzer (DMA), assuming ~10% of the introduced particles are selected as monodisperse aerosol particles."

***„Figure 1: what is the rationale behind the numbering.. why does (2) jump to pane d"***

This was due to the numbering of the logical steps of the components of PROTeGE.
We changed the numbering in Fig. 1.

***„Figure 4 there is no discussion at all why the time scale varies so much between panels a,b and c… Please discuss in the text what you want to show going from a) 1800sec to b) 450 sec to c) 20000 sec"***

This was due to the fact that some measurements were performed followed by each other without creating a new logging file on the OPS.
We did now separate all measurements and took only the first 420 seconds of all measurements and created new figures (Fig. 3 and Fig. 4) with equal length timescales.
For the comparison experiment between the printed and the brass orifice the data was measured in a single file to show the differences. Therefore, we also created a new figure (Fig. 5).

**„Figure 4 and 5. I suggest that the top panels with number concentrations should present the same extent of range… To have a visual meaning"**

We adjusted the top panels of Fig. 4 and Fig. 5 to the same range.

**„Right now the resolution is so poor in the figures that the legend of the top panels (fig4 and 5) are hardly readable."**

We adjusted the resolution for all figures in the manuscript to be better readable.

---

## Author Comment (AC2) · 15 Oct 2020

Authors' response

Dear Referees,

Thank you for your comments to improve the manuscript.
We have addressed your points and made the following changes to the manuscript.
We believe that the manuscript is greatly improved and is now ready for publication in AMT.

Thank you.

Best regards,

Michael Rösch and Dan Cziczo

**Referee 3:**

*„Is the technique improved over existing technology? The authors do not show data for D < 300 nm, the authors do not quantify the total number concentration of drops or typical droplet size produced, the authors do not quantify the composition of particles produced, the range of solvents that can be used (I guess some organic solvents might be problematic), the degree to which mixed particles (e.g. ammonium sulfate + organic compounds and preserving the ratio in the atomized particles) can be generated from*
*aqueous stock solutions, or the minimum aerosol diameter that can be generated, which is determined by cleanliness of the solvent and drop size, the maximum time the instrument can run unattended, the degree of drying that is needed, and the range of pressure and flow rates at which the atomizer produces particles. All of these are critical to evaluate if such a device is suitable for application in laboratory research, including for instrument calibration. Thus, the answer to the question is no."*

Due to the OPS detection limit of 0.3 $\mu$m we were not able to show data D < 300nm.
PROTeGE can run as long as there is solution to be dispensable and an air flow is present to disperse the droplets. The maximum time is limited by the

lifetime of the dryer downstream to ensure that the generated droplets are dried correctly.
The flow rate of PROTeGE is stated in the text.

*„Experiments including an SMPS to measure the full size distribution should be included. Experiments should systematically characterize the output for a much wider range of inputs (solvent, composition, solute weight percent) and analyze the results to infer drop number size and concentration. Ideally composition measurements of mixed particles are included to test for artifacts such as dissolution of the plastic and faithful representation of stock solution (e.g. adsorption of organics while the liquid passes*
*through the atomizer)."*

We regret that this is beyond the scope of this study. Certain solvents are not suitable for 3D printed parts. Since those are material specific the end user needs to make sure based on the MSDS to ensure proper functionality of the printed part.
As suggested, we performed a measurement with a SEMS to obtain particle number size distributions from 10 nm to 1000 nm for the generated ammonium sulphate solution of 0.6 g/L. We added the following paragraph to the manuscript followed by plot showing the average particle number size distribution of the generated ammonium sulfate particles (Fig. 4d).

"For the 0.6 g $L^{-1}$ solution an additional experiment using the SEMS instrument was performed. The size range was scanned from 10 nm to 1000 nm with a resolution of 60 bins and a sampling rate of 1 second per bin. The maximum particle number concentration was found at ~50 nm with ~40,000 $cm^{-3}$. The average PNSD for a 420 second sampling period is shown in Fig. 4d. During the experiment the generated size distributions did not change over time. Combining the obtained size distributions from SEMS and OPS shows that PROTeGE is capable of generating particles as small as 10 nm up to 2.4 $\mu$m based on the dispersed ammonium sulfate solution."

*„Does the work increase accessibility of the technology? The paper states that the authors were able to build this device, which is nice. However, there is no benefit to the community if it is not widely shared on how to do that. The authors state that STL files are "available upon request". This is insufficient. In my experience, share requests are often conveniently ignored or come with strings attached by the sharer. They present an unnecessary barrier. Thus, the answer to the question is no."*

Please see related comments by Referee #2: The .stl files are on a public repository for free download: https://www.thingiverse.com/thing:4444498

On the repository there are also pictures of PROTeGE and post processing details, this is now stated in the paper. Since the discussion paper was posted on AMTD we had more than 10 requests for the .stl file and already 16 downloads of the files from the repository.
The download statistics of the .stl file can also be found in the repository.

*„If the authors want this instrument to be a low cost, self-manufacture replacement, the authors should provide the STL files as a supplement or make them available in an archived repository. The paper should include an itemized list what people need to purchase, including part numbers and cost estimates. A photo of the instrument would be a good addition to the paper. The printing could be performed by a 3D printing service and ordered with a couple of clicks. Quotes can be generated from online vendors within minutes (e.g. sculpteo) by uploading the STL file. Assembly instruction should be provided. Comments about alternative print materials should be made and the precision that is needed for printing (is 100 micron the limit?). All of the designed parts should be made available using open licenses, e.g. the CERN open hardware license (https://www.ohwr.org/cernohl). Such a device would be very welcome and provide a platform where anyone could build, try, and characterize the output for themselves. In this case, the likely performance limitations and/or deficits in characterization raised earlier are less critical"*

Please see the answers above, this information is now provided and has been used by >10 readers of the discussion paper.

*„Irrespective the route the authors wish to pursue, the authors need to comment on the technical limitations above in the revised paper. The authors should also compare cost and performance to other techniques. For example, the TSI atomizer is $3k and very stable, and very well characterized. Small medical nebulizers (pressure and ultrasonic) can be obtained for < $30 and are more than sufficient to generate good aerosol for shorter duration (5-15 min). It might be useful to juxtapose data from these side-by-side and discuss use cases for the printed design."*

Please see the response to referee #2 first comment on the production cost for a PROTeGE generator. We did run an exhaust time experiment with an 80ml ammonium sulphate solution where after ~10 hours the nitrogen supply did run

out even before the solution did run out. Therefore, we conclude that PROTeGE is also capable of long-term production of aerosol.

We believe we have detailed the production and instrument performance within the paper so that potential users can compare this to other options. We in no way suggest PROTeGE should replace either high-end TSI atomizers nor nebulizers but, as has been demonstrated by multiple groups now using this technology, some researchers will find it the best solution for their needs.